# G6PD deficiency in Malaysia's Proto-Malay Orang Asli indigenous population: A molecular and epidemiological study

**Mohamed Afiq Hidayat Zailani**[1], **Raja Zahratul Azma Raja Sabudin**[1*],
**Muhammad-Redha Abdullah-Zawawi**[2], **Azlin Ithnin**[1], **Hafiza Alauddin**[1],
**Siti Aishah Sulaiman**[2], **Najiah Ajlaa Ayub**[1], **Rinie Awai@Albert**[1],
**Muhammad Dinie Afif Jamaludin**[1], **Muhammad Ziqrill Mohd Zapri**[1], **Ainoon Othman**[3]

1 Department of Pathology, Faculty of Medicine, Hospital Canselor Tuanku Muhriz, Universiti Kebangsaan Malaysia (UKM), Kuala Lumpur, Malaysia, 2 UKM Medical Molecular Biology Institute (UMBI), Universiti Kebangsaan Malaysia (UKM), Kuala Lumpur, Malaysia, 3 Department of Pathology, Faculty of Medicine and Health Sciences, Universiti Sains Islam Malaysia, Nilai, Negeri Sembilan, Malaysia

* zahratul@hctm.ukm.edu.my

## Abstract

Glucose-6-phosphate dehydrogenase deficiency (G6PDd) is one of the most common genetic disorders worldwide and remains highly prevalent in malaria-endemic regions. Individuals with G6PDd are at risk of severe complications, including acute haemolytic anaemia, when exposed to oxidative triggers. In Malaysia, the Proto-Malay Orang Asli (PMOA), the second largest indigenous group in Peninsular Malaysia, represents a vulnerable population. This study aimed to estimate the prevalence and mutation spectrum of G6PDd in this community. A total of 258 peripheral blood samples (91 males, 167 females) were screened using a quantitative G6PD assay (OSMMR2000-D). DNA from 73 samples was genotyped with the Hybribio G6PD GenoArray test, and 39 underwent targeted sequencing. The adjusted male median (AMM) of G6PD activity was 9.6 U/gHb (95% CI: 8.9–10.3 U/gHb), with 30% and 80% thresholds corresponding to 2.9 and 7.7 U/gHb, respectively. At the 30% cut-off threshold, the overall estimated prevalence of G6PDd was 6.8% (16/237; 12 males and 4 females). A total of 21 subjects were G6PD-intermediate (7 males and 14 females), and the remaining 221 subjects were G6PD-normal (72 males and 150 females). Genotyping identified 18 hemizygous males, 13 heterozygous females, and 3 homozygous females. Five G6PD variants were detected: G6PD Viangchan (39.5%), G6PD Coimbra (28.9%), G6PD Union (23.7%), G6PD Kaiping (5.3%), and rs782038151 (2.6%). This study demonstrates that G6PDd is common in the PMOA population, with notable molecular diversity. These findings have important implications for malaria control and the safe use of antimalarial drugs in this high-risk community.

**Data availability statement:** All relevant data are within the manuscript and its Supporting Information files.

**Funding:** RZA received the award that funded this study. This study was funded by the Ministry of Higher Education (MOHE), Malaysia through the Fundamental Research Grant Scheme (Grant number: FRGS/1/2020/SKK0/UKM/01/4). The website of funder is https://www.mohe.gov.my/. The funders had no role in study design, data collection and analysis, decision to publish, or preparation of the manuscript.

**Competing interests:** The authors have declared that no competing interests exist.

## 1. Introduction

Glucose-6-phosphate dehydrogenase deficiency (G6PDd) is a hereditary condition caused by loss-of-function mutations of the G6PD gene on the X-chromosome (Xq28). The mutations result in the disruption of protein folding and alteration of protein structure, which leads to decreased enzyme activity [1]. Hemizygous males carrying the mutant allele will show full phenotypical expression, i.e., deficient in the enzyme. Affected females may either be homozygous or compound heterozygous when the mutant alleles are carried on both X chromosomes or heterozygous when the individual carries the mutant allele on one X chromosomes with the other X-chromosome being wild type.

In humans, G6PD is essential for cellular metabolism and defense against oxidative stress. It generates NADPH by regulating glucose metabolism through the hexose monophosphate (HMP) pathway, where $NADP^+$ is reduced to NADPH. [2]. In red blood cells (RBCs), this pathway is the only source for NADPH production, a key substance in maintaining a high level of reduced glutathione, which protects RBCs against oxidative damage [3]. Upon exposure to oxidative substances, the RBCs in G6PD-deficient patients are unable to generate sufficient NADPH, resulting in destruction of RBCs and severe acute hemolytic anaemia.

The prevalence of G6PD deficiency have been evaluated globally since it was first identified by researchers in 1956 [4]. According to the World Health Organization (WHO), the estimated frequency of G6PD deficiency carriers and G6PD-deficient individuals among the global human population are 7.5% and 2.9%, respectively [5]. Further assessments show that the frequency of G6PD deficiency varies significantly among groups, ethnicities, and regions. High frequencies of G6PDd were also seen in areas where malaria is endemic [6]. Based on a geostatistical model-based map, the sub-Saharan African region was reported to have the highest G6PDd prevalence, followed by the Arabian Peninsula, Central and Southeast Asia, Mediterranean Europe, and Latin America [7]. Studies demonstrated that Asian countries, such as Malaysia, Indonesia, and Thailand, account for a significant portion of G6PD-deficient individuals worldwide due to the region's high population.

In Malaysia, G6PDd is a major public health concern, and the frequency distribution of G6PD deficiency differs across diverse ethnicities [8]. Historically, G6PDd was found in earlier studies to be more prevalent among Malays and Malaysian Chinese, and less common among Malaysian Indians [9–12]. A most recent study showed an overall prevalence of G6PD deficiency of 3.4% among the general population in Malaysia, with the Malays showing a prevalence of 4.6% in males and 1.3% in females, the Chinese 7.2% in males and 0.7% in females, and the Indian 2.7% in males and 0.7% in females, respectively [13].

The Malaysian indigenous populations, known as the Orang Asli, consisted of three large groups, namely the Negrito, Senoi, and Proto-Malay. Their remote locations leads to multiple significant health issues, including infections such as malaria, malnutrition, and poor hygiene practices. Previous studies have examined malaria endemicity among the Malaysian Orang Asli. In 2025, a study of 437 Orang Asli reported a 15.3% prevalence of malaria, with *Plasmodium vivax* being the most

common species (8.7%, 38 of 437) [14]. Past studies also showed that there were high G6PDd prevalences among the two Orang Asli ethnic groups, the Negrito and the Senoi (9%, and 15.2%, respectively), with their molecular variants have been characterized [15,16]. To date, no studies on G6PDd have been conducted among the Malaysian Proto-Malay Orang Asli (PMOA) group. This study therefore aims to determine the estimated prevalence of G6PDd and its molecular variants within the PMOA population. Understanding G6PDd across different Malaysian ethnicities is crucial for reducing the national disease burden and supporting policymakers in developing effective strategies for malaria eradication. In Malaysia, Primaquine, an 8-aminoquinoline, is prescribed for the radical cure of *Plasmodium vivax* and as a gametocytocidal agent. As Primaquine can trigger hemolysis in G6PD-deficient individuals, determining its prevalence is clinically important to ensure safe malaria treatment.

## 2. Materials and methods

### 2.1 Study design

A cross-sectional study was conducted to assess G6PDd in the PMOA population in Peninsular Malaysia. The ethical approval was obtained from the Universiti Kebangsaan Malaysia (UKM) Medical Centre Ethics Committee (Reference Number: UKM PPI/111/8/JEP-2021–069). Participant recruitment for this study was conducted over nine months, from June 26th, 2022, to March 12th, 2023.

A total of 258 individuals from the PMOA group were enrolled in this study, including 91 males and 167 females. The participants were selected from the PMOA community residing in the west coast states of Peninsula Malaysia, namely Selangor and Negeri Sembilan, as well as Peninsula Malaysia's east coast state of Pahang. These states constitute the main settlement areas of the PMOA group, according to the Malaysian Department of Orang Asli Development (JAKOA) demographic records. Logistical feasibility and accessibility of the communities also guided our sampling. The inclusion criteria include healthy PMOA individuals with pure, single subethnic lineage for three generations. Participants with an unclear or mixed subethnic lineage and subjects with any diagnosed systemic or blood disease were excluded from our study. Each participant underwent interviews and screenings.

### 2.2 Inform consent

The participants were given full explanations on the objectives and background of our study. Then, a written consent was obtained from each subject before collecting their blood samples. For participants aged under 18 years old, written consent was obtained from their parent or legal guardian. The flow chart summarizes the study design and laboratory analyses, including gender distribution and participant numbers at each stage (Fig 1).

### 2.3 Sample collection and processing

A total of 7 ml of venous blood samples were collected from each participant in ethylene diamine tetraacetic acid (EDTA) tubes using sterile techniques. The collected blood samples were transported to the laboratory at the Hematology Unit, Department of Diagnostic Laboratory Services, Hospital Canselor Tuanku Mukhriz (HCTM) within six hours post-collection. The blood samples were subjected to a full blood count (FBC) analysis and quantification of G6PD activity using a spectrophotometric assay kit, OSMMR2000-D assay kit with hemoglobin (Hb) normalization (OSMMR; R&D Diagnostics, N. Dimopoulos S.A, Greece).

The OSMMR2000-D assay protocol comprised multiple sequential steps. A total of 5 µL of the collected blood samples were added into the wells of a 96-well microplate using an Eppendorf® Research® plus, adjustable 1-channel micropipette sized 2–20 µL (Eppendorf Inc., Germany). The blood samples were mixed with 75 µL of elution buffer that was provided along with the test kits, in the microplate wells. The microplate was placed on the Awareness STAT FAX® 2200 Digital Incubator Shaker (Awareness Technology, Florida, USA) and was warmed at 37°C. On another microplate, 75 µL of the

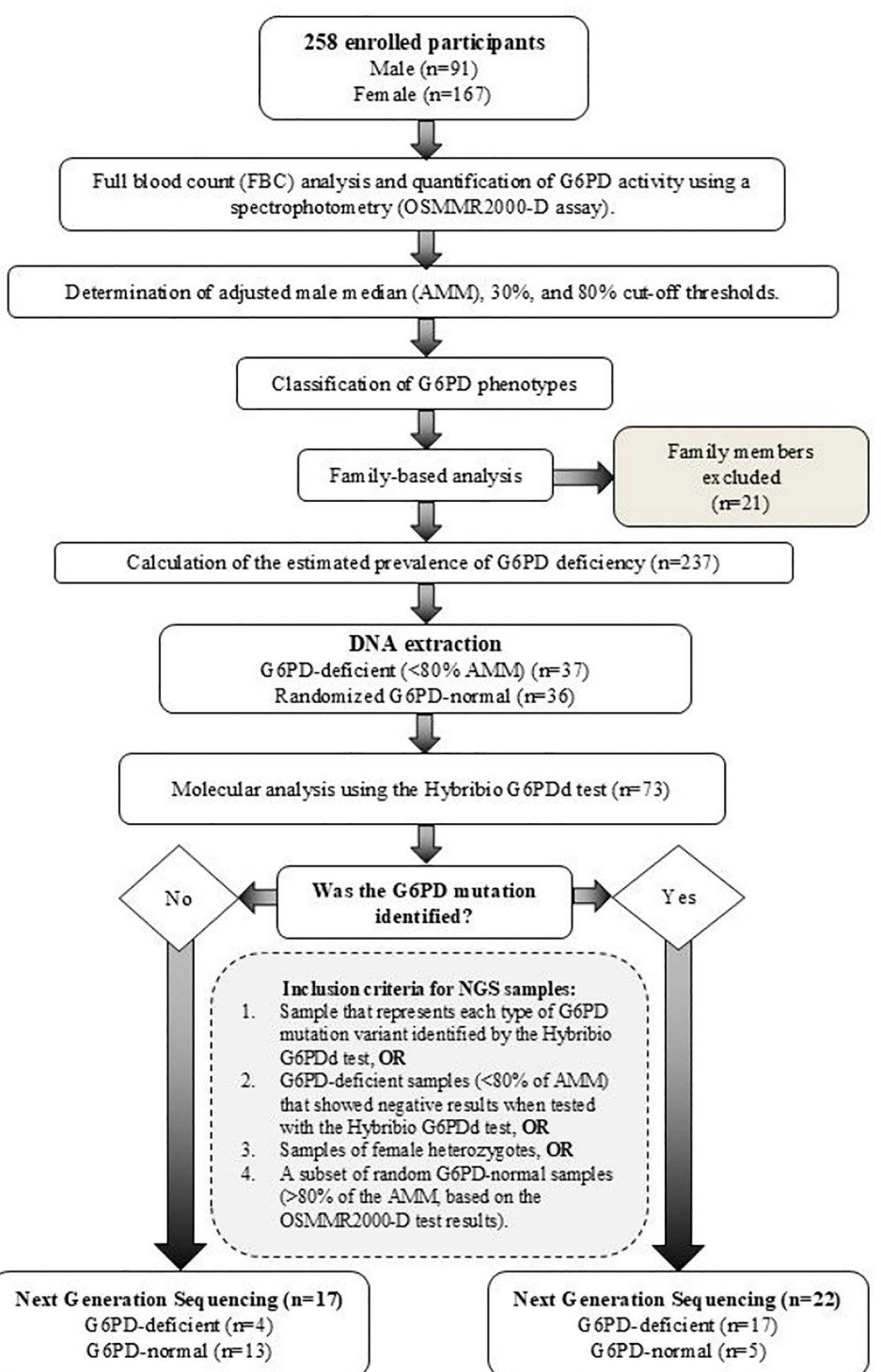

**Fig 1. Flow chart summarizing the study design and laboratory analyses.**

OSMMR2000-D test reagent was added to the wells and was warmed to a similar experiment temperature of 37°C. The first microplate that was incubated in the shaker was taken out, and a total of 15 μL of each eluted sample was transferred to the corresponding well on the second microplate that contained the reagent and was mixed well.

The first reading of the microplate was taken at a wavelength of 405 nm for Hb measurement using a BioTek® ELx808™ Absorbance Plate Reader (Agilent Technologies Inc., Santa Clara, USA). The wells of the microplate were filled with 100 μL of the color reagent mixture provided in the OSMMR2000-D test kit. The reader was set to an "endpoint" mode, and the microplate was read for the second and third times at a wavelength of 550 nm, at 0 min and 10 min, respectively. G6PD activity results produced from the OSMMR2000-D assay were directly expressed in U/gHb and generated through the WinKQCL™ Endotoxin Detection & Analysis Software version 6.3.

The adjusted male median (AMM) of the G6PD activity results was determined for the PMOA population. The AMM was then defined as 100% G6PD activity, as commonly implemented in prior G6PD research [16–18]. The AMM calculation was performed as follows: the median G6PD activity of all male participants was calculated, samples with less than 10% of the median activity were eliminated, and the median of the remaining samples (i.e., samples >10% of the median) was recalculated. Consistent with this AMM definition, we did not genotype-filter males with normal enzyme activity (including those carrying synonymous or intronic variants such as C1311T or T93C) [17].

The 30% and 80% cut-off thresholds of the AMM were then determined to classify the G6PD phenotypes of the participants according to the latest WHO recommendations [19]. They were identified as G6PD-deficient if their enzyme levels were below 30% of the AMM, G6PD-intermediate if their enzyme levels were between 30% and 80% of the AMM, and G6PD normal if their enzyme levels were beyond 80% of the AMM. Although the G6PD-intermediate category (30–80% of the AMM) was used mainly for females due to clinical relevance, for our research analysis, we have included the G6PD-intermediate category for the male subjects.

## 2.4 Molecular analysis

DNA extractions and molecular analyses of all G6PD-deficient samples with enzyme levels below 80% of the AMM (n = 37) and a randomized subset of G6PD-normal samples (n = 36) were conducted using a Reverse Dot-Bot Flow-Through Hybridization (RDB-FTH) technique, via the Hybribio G6PDd Genoarray Test Kit (Hybribio G6PDd; Chaoxhou Hybribio Biochemistry Ltd., Sheung Wan, Hong Kong) to characterize G6PD mutations. Briefly, the Hybribio G6PDd test was designed to detect the genotypes of G6PD mutation variants, using a developed panel of 14 known G6PD mutation variants in Malaysia, namely G6PD Viangchan (871 G > A), G6PD Kaiping (1388 G > T), G6PD Mediterranean (563 C > T), G6PD Mahidol (487 G > A), G6PD Canton (1376 G > T), G6PD Union (1360 C > T), G6PD Coimbra (592 C > T), G6PD Vanua Lava (383 T > C), G6PD Gaohe (95 A > G), G6PD Chinese-5 (1024 C > T), G6PD Orissa (131 C > G), G6PD Chatham (1003 G > A), G6PD Andalus (1361 G > A) and G6PD Quing Yang (392 G > T).

Following the guidelines provided by the developer, the results of the Hybribio G6PDd test were interpreted based on the presence of colored dots on the specific probes created on the MEM-G6PD membrane. Homozygous or hemizygous samples show dots on the mutant (M) probes; heterozygous samples show dots on both the mutant (M) and normal (N) probes; while normal or undetected samples show dots only on all normal (N) probes. Control samples display dots on the biotin control spot. (Source: Instruction Protocol of Hybribio G6PD Diagnostic Test Kit, Version 1.1) (Fig 2).

Following the molecular analysis by the Hybribio G6PDd test, a total of 39 samples were further sequenced via targeted sequencing (Next-Generation Sequencing; NGS). The inclusion criteria for these samples include the following: the sample that represents each type of G6PD mutation variant identified by the Hybribio G6PDd test, samples with G6PD activities <80% of the AMM that showed negative results when tested with the Hybribio G6PDd test, samples of females heterozygotes, and a subset of random G6PD-normal samples (>80% of the AMM). The selected samples were required to fulfill one or more of these criteria. The objective of this method was threefold: to validate the results from the Hybribio G6PDd test, determine its concordance rate, and detect any additional or previously uncharacterized

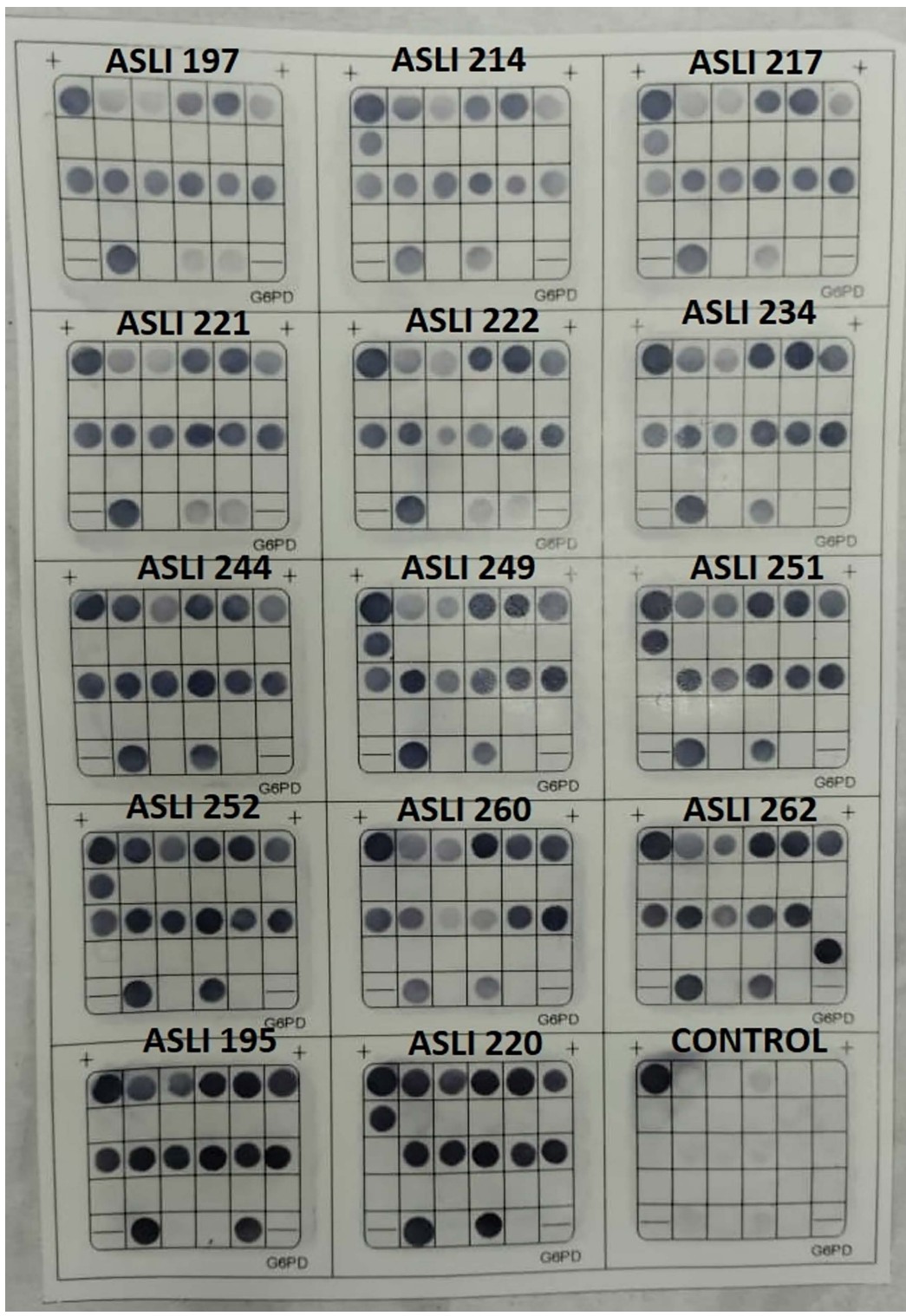

**Fig 2. Example of the result obtained on the MEM-G6PD membrane of the Hybribio G6PDd test.** Results were interpreted based on colored dots appearing on specific probes of the MEM-G6PD membrane: mutant (M) probes for homozygous/hemizygous samples, both M and normal (N) probes for heterozygous samples, and only N probes for normal/undetected samples. A biotin control spot is shown in the bottom box (far right). The exact positions of these dots, and the mutations they represent, are provided in the kit protocol.

G6PD mutation variants among the PMOA population. The targeted sequencing was performed at the coding sequence (CDS region) of the G6PD gene. The procedure was fully conducted at an external genomic service laboratory of Analisa Resources (M) Sdn. Bhd, Selangor, Malaysia, the sole distributor for QIAGEN academic research and applied testing portfolios. The procedures include extracted DNA sample receipt, initial purity Quality Control (QC) check which was analyzed using the QIAseq targeted DNA Pro Custom Panel -CPHS-46443Z-27 (Qiagen, Germany), library construction using QIAseq Targeted DNA Pro Custom Panel (QIAGEN, Hilden, Germany) and QIAseq Targeted DNA Pro UDI Set A (QIAGEN, Hilden, Germany), QC for libraries, and targeted sequencing using Illumina Novaseq 6000 system platform (Illumina, CA, USA).

## 2.5 Statistical data analysis

Participants' data, including their G6PD activities and G6PD status, were collated using Microsoft 365 Excel Spreadsheet Software (Microsoft Corporation, WA, USA). The G6PDd prevalence in the PMOA group was calculated after excluding all related family members. The G6PD results were statistically analyzed using IBM SPSS Statistics 26.0 for Windows. The molecular results of the Hybribio G6PD test, including the allelic frequencies of all identified variants, were collated and analyzed against findings of previous studies that were conducted in other Malaysian populations.

## 2.6 Sequencing data analysis

For secondary data analysis of this study, sequencing reads were trimmed to remove adapters and low-quality bases using Trimmomatic (v0.30). The resulting high-quality reads were then aligned to the human reference genome GRCm38 using BWA-MEM (v0.7.17) in paired-end mode. Genome pileup generation and variant calling were performed using BCFTools (v1.12). For our tertiary analysis, variants were annotated and prioritized using established pipelines, including functional annotation via wANNOVAR and classification based on the American College of Medical Genetics guidelines. First, variants were filtered to include only those located in the G6PD gene. Second, variants with a population frequency $> 0.05$ based on the Global Allele Frequency from the Genome Aggregation Database (gnomAD), 1000 Genomes Project, and Exome Aggregation Consortium (ExAC) were excluded. Third, variants were prioritized based on their predicted functional impact, retaining nonsynonymous, stop-gain, stop-loss, splicing, frameshift, and non-frameshift insertion/deletion variants located in exonic and splicing regions for further analysis. Lastly, candidate variants predicted as likely benign or benign were excluded using computational tools such as SIFT, PolyPhen, and MutationTaster. However, other regions, such as 3'UTR and intronic, were also observed in this study, which could play a crucial role in regulating mRNA stability, translation, and localization of G6PD.

## 3. Results

### 3.1 G6PD activity analysis

A total of 258 participants from the PMOA ethnicity were enrolled in this study, including 91 males (35.3%) and 167 females (64.7%). A total of 40 participants (15.5%) were children aged 4–12 years old and 218 subjects (84.5%) were adults aged between 13–70 years old. The AMM of the PMOA population was 9.6 U/gHb (95% CI, 8.9–10.3 U/gHb). The 30% and 80% cut-off values of the AMM G6PD activities corresponded to 2.9 U/gHb and 7.7 U/gHb, respectively.

A total of 21 out of 258 (8.1%) participants were identified to be family members and were excluded from the calculations of G6PDd prevalence. At the 30% cut-off threshold, the overall estimated prevalence of G6PDd was 6.8% (16/237; 12 males and 4 females). A total of 21 subjects were G6PD-intermediate (7 males and 14 females), and the remaining 221 subjects were G6PD-normal (72 males and 150 females). The levels of G6PD activity among the PMOA participants are detailed in the study's minimal underlying data set (S1 file).

In male participants, the range of G6PD activities for each phenotypic classification was 1.1–2.4 U/gHb (G6PD-deficient), 2.9–7.6 U/gHb (G6PD-intermediate), and 7.8–14.0 U/gHb (G6PD-normal). Meanwhile, in female participants,

the range of G6PD activities for each phenotypic classification was 0.1–2.7U/gHb (G6PD-deficient), 3.6–7.6 U/gHb (G6PD-intermediate), and 7.7–20.8 U/gHb (G6PD-normal) (Table 1).

### 3.2  G6PD mutation analysis by the Hybribio test

A total of 73 extracted DNA samples of the PMOA population (73/258; 28%) were analyzed using an RDB-FTH technique, via the Hybribio G6PDd test. Based on the spectrophotometric OSMMR2000-D test, the DNA include 37 samples (19 males and 18 females) that had G6PD activity <80% of AMM, while the remaining 36 samples (17 males and 19 females) were G6PD-normal (>80% of AMM). When tested using the Hybribio G6PDd test, 33 of the 37 deficient samples showed the presence of G6PD mutation, and 5 of the 36 G6PD-normal samples showed the presence of G6PD mutation.

Overall, four G6PD mutations were found by the Hybribio G6PDd test with following frequencies: G6PD Viangchan (c.871G>A) (5.4%; n=14), G6PD Coimbra (c.592C>T) (4.7%; n=12), G6PD Union (c.1360C>T) (4.3%; n=11), and G6PD Kaiping (c.1388G>A) (0.8%; n=2). The cases with G6PD Viangchan include seven hemizygous males, five heterozygous females, and two homozygous females. The G6PD Coimbra cases include four hemizygous males, six heterozygous females, and one homozygous female. The G6PD Union cases include four males and six heterozygous females. The G6PD Kaiping case was found in one hemizygous male and one compound heterozygous female (combined G6PD Coimbra/Kaiping).

The five normal samples with the presence of G6PD mutations include G6PD Viangchan (one heterozygous female; G6PD activity based on the OSMMR2000-D test: 83% of the AMM), G6PD Coimbra (two heterozygous females; G6PD activity based on the OSMMR2000-D test: 83% and 85% of the AMM), and G6PD Union (two heterozygous females; G6PD activity based on the OSMMR2000-D test: 81% and 89% of the AMM). The remaining 35 samples (4 deficient and 31 normal) revealed negative results (normal or undetected cases). All four deficient samples with negative results (two males and two females) had intermediate G6PD activities (75% and 79% of the AMM for males; 78% and 79% of the AMM for females) (Fig 3 and Table 2).

### 3.2  G6PD mutation analysis by the targeted gene sequencing

The characteristics of the 39 selected samples analyzed using NGS were as follows: 22 samples (17 samples <80% AMM and 5 G6PD-normal) that showed the presence of G6PD mutations in the Hybribio G6PDd test, and 17 samples (4 samples <80% AMM and 13 G6PD-normal) that showed normal/undetected mutations, i.e., negative results. The NGS demonstrated that 38 of the 39 samples were correctly identified by the Hybribio G6PDd test, resulting in a 97.4% concordance rate with the NGS. A sample labeled ASLI 040 was found to have G6PD Coimbra (c.592C>T), which showed a negative result (normal/undetected mutation) by the Hybribio G6PDd test, and it belonged to a 29-year-old heterozygous female with G6PD activity at 84% of the AMM. For the 4 deficient samples (2 males and 2 females; samples labeled ASLI 046, ASLI 052, ASLI 054, and ASLI 070) that showed normal/undetected mutation by the Hybribio G6PDd test, NGS found no

**Table 1. Frequency of cases and range of G6PD activities measured by the OSMMR2000-D test (n=258).**

| Gender | G6PD-Deficient (<30% of AMM) | | G6PD-Intermediate (30–80% of AMM) | | G6PD-Normal (>80% of AMM) | | Total |
|---|---|---|---|---|---|---|---|
| | Number of cases (N) | Range of G6PD activity (U/gHb) | Number of cases (N) | Range of G6PD activity (U/gHb) | Number of cases (N) | Range of G6PD activity (U/gHb) | |
| Male | 12 | 1.1–2.4 | 7 | 2.9–7.6 | 72 | 7.8–14.0 | 91 |
| Female | 4 | 0.1–2.7 | 14 | 3.6–7.6 | 150 | 7.7–20.8 | 167 |
| Total | 16 | 0.1–2.7 | 21 | 2.9–7.6 | 221 | 7.7–20.8 | 258 |

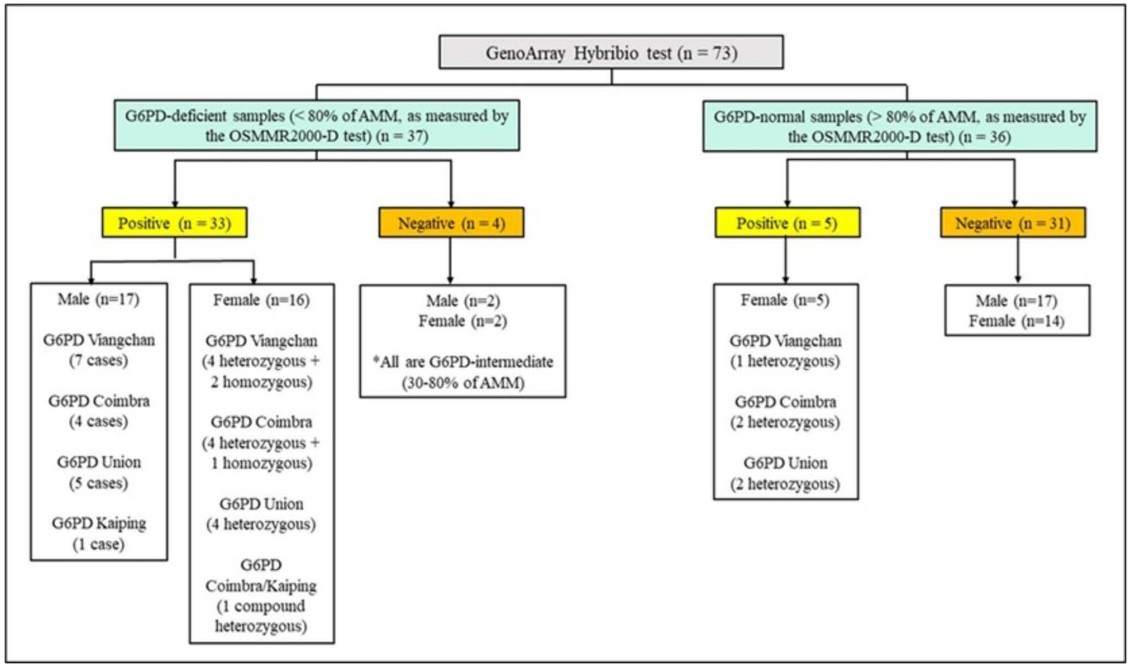

**Fig 3. Tree diagram showing the results of the Hybribio G6PDd test.** A total of 38 samples exhibited mutation variants: G6PD Viangchan (c.871G>A) (14 cases), G6PD Coimbra (c.592C>T) (12 cases), G6PD Union (c.1360C>T) (10 cases), and G6PD Kaiping (c.1388G>A) (2 cases).

G6PD mutation, and based on the OSMMR2000-D test, they were identified as G6PD-intermediate (G6PD activity: 75% and 79% of the AMM for males, 78% and 79% of the AMM for females.

NGS analysis found no unknown or novel exonic G6PD mutation in the PMOA population. Interestingly, one sample labeled ASLI 262, which was confirmed to harbor G6PD Kaiping (c.1388G>A), was identified with an additional G6PD mutation, rs782038151 (g.154535364T>G), a known yet rare missense variant. The sample belonged to a 46-year-old hemizygous male who had G6PD activity of 3.2 U/gHb (33% of the AMM). A total of 28 samples with the C1311T synonymous mutation in exon 11 were found, and only one with combination of G6PD Viangchan (c.871G>A). Intronic mutations of the G6PD gene were found in all 39 samples. Among them, 30 samples showed a benign non-pathogenic single nucleotide variant (Variant ID: 982298) located at chrX:154532293(GRCh38) and one sample showed a benign non-pathogenic single nucleotide variant (Variant ID:1780243) located at chrX:154538857(GRCh38). The remaining eight samples showed unknown intronic variants (i.e., intronic variants with untranslated regions). A total of 35 out of 39 samples showed the presence of 3'-untranslated region (UTR) polymorphism, which is a non-coding region of the G6PD gene. Among them, five samples showed a missense mutation of the 1780209 (c.1792C>G) variant, while the remaining 30 samples showed 3'-UTR variants of uncertain significance (VUS). No cases with the T93C intronic variant were found in this study.

### 3.3 Allelic frequencies of G6PD mutations

Overall, 44 G6PD mutant alleles were detected from 40 PMOA samples (34 samples <80% AMM and six G6PD-normal). Among the 34 deficient samples, 18 samples were hemizygous males, 13 samples were heterozygous females, and three samples were homozygous females. The allelic frequencies of G6PD mutations among the samples with <80 AMM were as follows: G6PD Viangchan (c.871G>A) (39.5%), G6PD Coimbra (c.592C>T) (28.9%), G6PD Union (c.1360C>T)

**Table 2. G6PD mutation variants detected by the Hybribio G6PDd test among the Proto Malay Orang Asli population (n = 73).**

| Variant of Mutation | Number of Case | Phenotype* (n) | Genotype | Gender | G6PD Activity (U/gHb)** | | | |
| --- | --- | --- | --- | --- | --- | --- | --- | --- |
| | | | | | Range | Mean (±SD) | 95% CI | Median |
| G6PD Viangchan (c.871G>A) | 14 | G6PD-deficient (4) | Hemizygote | Male | 2.0–2.8 | 2.3±0.4 | 1.6–2.9 | 2.3 |
| | | G6PD-intermediate (3) | Hemizygote | Male | 3.1–4.2 | 3.7±0.6 | 2.3–5.1 | 3.8 |
| | | G6PD-deficient (2) | Homozygote | Female | 2.5–2.7 | 2.6±0.1 | 1.3–3.9 | 2.6 |
| | | G6PD-intermediate (4) | Heterozygote | Female | 3.7–7.1 | 6.0±1.5 | 3.5–8.4 | 6.5 |
| | | G6PD-normal (1) | Heterozygote | Female | 8.0 | | | |
| G6PD Coimbra (c.592C>T) | 11 + 1[a] | G6PD-deficient (4) | Hemizygote | Male | 1.1–1.3 | 1.2±0.1 | 1.0–1.3 | 1.1 |
| | | G6PD-deficient (1) | Homozygote | Female | 0.1 | – | – | – |
| | | G6PD-deficient (1[a]) | Compound Heterozygote | Female | 1.7 | – | – | – |
| | | G6PD-intermediate (4) | Heterozygote | Female | 4.7–7.3 | 6.5±1.2 | 4.6–8.4 | 6.9 |
| | | G6PD-normal (2) | Heterozygote | Female | 8.0–8.2 | 8.1±0.1 | 6.8–9.4 | 8.1 |
| G6PD Union (c.1360C>T) | 11 | G6PD-deficient (5) | Hemizygote | Male | 0.9–1.5 | 1.3±0.3 | 1.0–1.7 | 1.4 |
| | | G6PD-intermediate (4) | Heterozygote | Female | 3.6–7.6 | 5.5±1.7 | 2.8–8.1 | 5.3 |
| | | G6PD-normal (2) | Heterozygote | Female | 7.8–8.6 | 8.2±0.6 | 3.1–13.2 | 8.2 |
| G6PD Kaiping (c.1388G>A) | 1 + 1[a] | G6PD-intermediate (1) | Hemizygote | Male | 3.2 | – | – | – |
| | | G6PD-deficient (1[a]) | Compound Heterozygote | Female | 1.7 | – | – | – |
| Normal or Undetected Variant (Negative results) | 35 | G6PD-intermediate (2) | | Male | 7.2–7.6 | – | – | – |
| | | G6PD-intermediate (2) | | Female | 7.5–7.6 | – | – | – |
| | | G6PD-normal (17) | | Male | 7.8–11.7 | – | – | – |
| | | G6PD-normal (14) | | Female | 7.8–10.7 | – | – | – |

(23.7%), G6PD Kaiping (c.1388G>A) (5.3%), and rs782038151 (2.6%). For the normal samples, all six were heterozygous females. The allelic frequencies of G6PD mutations among the normal samples were as follows: G6PD Coimbra (c.592C>T) (50.0%), G6PD Union (c.1360C>T) (33.3%), and G6PD Viangchan (c.871G>A) (16.7%) (Tables 3 and 4). A consolidated supplementary table presents all detected mutations with their allele frequencies, including variants in intronic and untranslated regions (S2 File).

**Table 3. Allelic frequencies of G6PD mutations among the Proto Malay Orang Asli deficient and intermediate samples (<80% of AMM) (n = 34).**

| G6PD Mutation Variant | Male (N) | Hetero-zygous female (N) | Homo-zygous female (N) | Compound heterozy-gous females (N) | Calculation for the Total mutant allele (N) | Allelic Frequency |
| --- | --- | --- | --- | --- | --- | --- |
| G6PD Viangchan (c.871G>A) | 7 | 4 | 2 | 0 | 7+4+(2x2)+0 = 15 | 39.5% (15/38) |
| G6PD Coimbra (c.592C>T) | 4 | 4 | 1 | 1 | 4+4+(1x2)+1 = 11 | 28.9% (11/38) |
| G6PD Union (c.1360C>T) | 5 | 4 | 0 | 0 | 5+4+0+0 = 9 | 23.7% (9/38) |
| G6PD Kaiping (c.1388G>A | 1 | 0 | 0 | 1 | 1+0+0+1 = 2 | 5.3% (2/38) |
| rs782038151 | 1 | 0 | 0 | 0 | 1+0+0+0 = 1 | 2.6% (1/38) |
| Total | 18 | 13 | 3 | 2 | 38 | 100% |

**Table 4. Allelic frequencies of G6PD mutations among the Proto Malay Orang Asli normal samples (>80% of AMM) (n = 6).**

| G6PD Mutation Variant | Heterozygous female (N) | Total mutant allele (N) | Allelic Frequency |
|---|---|---|---|
| G6PD Coimbra (c.592C>T) | 3 | 3 | 50.0% (3/6) |
| G6PD Union (c.1360C>T) | 2 | 2 | 33.3% (2/6) |
| G6PD Viangchan (c.871G>A) | 1 | 1 | 16.7% (1/6) |
| Total | 6 | 6 | 100% |

## 4. Discussion

### 4.1 Prevalence and distribution of G6PD activity

The prevalence and distribution of G6PD activity in a studied population are commonly assessed using the AMM, which defined by WHO as "the median G6PD activity value in a target population of males who are not G6PD-deficient" [20,21]. However, there was a limited studies reported the AMM using the spectrophotometric OSMMR2000-D test, as past studies that used this test reported mean G6PD activity rather than the AMM [22–25]. Shifts in the methodological approaches from mean to AMM by researchers worldwide may have been driven by the need for more accurate representations of G6PD activity levels, as the mean could be skewed by extreme values in individuals with severe deficiencies. In this study, the AMM was 9.6 U/gHb (95% CI, 8.9–10.3 U/gHb). This value was lower than that reported among the Malaysian Senoi Orang Asli (11.8 U/gHb) [16]. However, it was slightly higher than that reported among Thailand's population using a similar test (8.1 U/gHb) [26].

Earlier research showed that the AMM differs greatly across 13 studies that used the Trinity spectrophotometric assay kit, ranging from 5.7 to 12.6 U/gHb [27]. The differences in the AMM between sites and studies might be due to several factors, including variations in laboratory methods and possible unmeasured demographic factors. Based on the AMM, our 30% and 80% cut-off corresponded to 2.9 U/gHb and 7.7 U/gHb, respectively. These values were close to those measured using the reference spectrophotometric assays in previous studies, ranging from 2.1 to 2.97 U/gHb for the 30% cut-off threshold and 5.5 to 7.2 U/gHb for the 80% cut-off threshold [26,28–31].

At a 30% cut-off threshold, the estimated prevalence of G6PDd among the PMOA population was 6.8% (16/237). This prevalence is lower than that reported in Malaysia's Senoi Orang Asli group (9.8%; 36/369) at the same cut-off threshold using the same quantitative method. It also falls within the range observed in the general Malaysian ethnic groups, including Malays, Malaysian Chinese, and Negrito Orang Asli (5.9–9.0%) [15,16,32]. Nonetheless, the prevalence should be interpreted with caution, as it may reflect differences in the employed G6PD testing method (i.e., quantitative G6PD activity assay), compared to most earlier studies that used the FST, a qualitative method that could have underestimated G6PD cases. Past study revealed that the G6PDd prevalence detected by the FST was significantly lower (2.12%; 5/236) than that detected by the quantitative G6PD assay (i.e., the OSMMR2000-D test), which was 9.32% (22/236) [23].

G6PDd among the PMOA population was also more prevalent in males than females. G6PDd is an X-linked recessive genetic condition, and the observed gender distribution aligns with patterns reported in previous studies [33,34]. The identification of G6PDd prevalence and distribution in this remote population has direct implications for malaria elimination programs, as the safe administration of 8-aminoquinoline drugs, including primaquine and tafenoquine, requires accurate identification of G6PD-deficient individuals. Implementing reliable point-of-care quantitative tests in these remote and high-risk areas could minimize the risk of drug-induced hemolysis and improve malaria treatment outcomes. Beyond malaria, our findings also support the inclusion of G6PDd in national newborn screening programs. Early identification of affected infants, particularly in such indigenous populations, could reduce neonatal morbidity from hemolytic crises and inform safe prescribing practices later in life. Additionally, the majority of the PMOA with this genetic condition were

G6PD-intermediate (30–80% of AMM)(21/37). The identification of a significant number of G6PD-intermediate activity through the spectrophotometric OSMMR2000-D test emphasizes the importance of employing quantitative G6PD activity assays for targeted population screening.

### 4.2 Spectrum of mutations, allelic frequencies, and genetic flows

Molecular characterization showed the presence of five variants among samples <80% AMM with varying allelic frequencies (Fa). Of these, three were dominant: G6PD Viangchan (c.871G>A) (Fa=39.5%), G6PD Coimbra (c.592C>T) (Fa=28.9%), and G6PD Union (c.1360C>T) (Fa=23.7%). The remaining two variants showed lower allelic frequencies: G6PD Kaiping (c.1388G>A) (Fa=5.3%) and rs782038151 (Fa=2.6%). Among the G6PD-normal subjects, the spectrum of mutations and allelic frequencies were G6PD Coimbra (c.592C>T) (Fa=50%), G6PD Union (c.1360C>T) (Fa=33.3%), and G6PD Viangchan (c.871G>A) (Fa=16.7%). These findings were consistent with past studies that showed a specific population commonly has two to three dominant variants within its unique spectrum [7]. The distribution of G6PD variants across the three major ethnic groups of the Malaysian Indigenous Orang Asli population demonstrates both shared and distinct patterns (Table 5).

G6PD Viangchan (c.871G>A) is the most common variant found in this study [35]. In the Southeast Asian region, this mutation is a dominant variant that causes the majority of G6PD-deficient cases [36]. Through these findings, it can be hypothesized that the PMOA group may have a strong historical connection with populations in Southeast Asian regions. In Malaysia, it is also dominant in the Malay, Negrito Orang Asli, and Senoi Orang Asli populations, but only in low frequency among the Malaysian Chinese population (0.8%) [15,16,32,37]. Previous studies have reported that G6PD Viangchan (c.871G>A) and two silent mutations, c.1311C>T in exon 11 and the T93C intronic variant (1311T/93C), are often co-inherited as part of specific haplotypes [38]. However, in our dataset, only a single sample carrying the G6PD Viangchan (c.871G>A) variant was also found to harbor the C1311T synonymous mutation, and no co-occurrence with the T93C intronic variant was observed. This observation could indicate that co-inheritance patterns in the PMOA population may differ from those reported in other Southeast Asian populations, although this finding should be interpreted cautiously given the limited sample size and potential population-specific genetic variation.

The second most common variant was G6PD Coimbra (c.592C>T) which has a wide distribution across Europe and Asia [39,40]. In Southeast Asia, earlier studies documented this variant in several countries, such as Indonesia, Myanmar, and Cambodia [41–43]. In Malaysia, low frequencies of G6PD Coimbra (592C>T) were reported in Malays, Negrito Orang

**Table 5. G6PD mutation variants among the three main groups of the Malaysian Orang Asli population.**

| Type of Malaysian Orang Asli group | G6PD mutation variants | Nucleotide substitution | WHO Class | Frequency (%) | Reference |
|---|---|---|---|---|---|
| Negrito Orang Asli | G6PD 1311T/93C<br>G6PD Viangchan<br>G6PD Coimbra | 1311C>T and<br>93T>C<br>871G>A<br>592C>T | C<br>B<br>B | 64<br>12<br>4 | Amini et al. 2011 |
| Senoi Orang Asli | G6PD 1311T/93C<br>rs1050757<br>G6PD Viangchan<br>G6PD Union<br>G6PD Kaiping<br>G6PD Coimbra | 1311C>T and<br>93T>C<br>357A>G<br>871G>A<br>1360C>T<br>1388G>A<br>592C>T | C<br>C<br>B<br>B<br>B<br>B | 47.1<br>39.1<br>25.3<br>21.8<br>8.0<br>2.3 | Danny et al. 2023 |
| Proto Malay Orang Asli | G6PD Viangchan<br>G6PD Coimbra<br>G6PD Union<br>G6PD Kaiping rs782038151 | 871G>A<br>592C>T<br>1360C>T<br>1388G>A<br>chrX:154535364 (T>G) | B<br>B<br>B<br>B | 36.3<br>31.8<br>25.1<br>4.5<br>2.3 | Zailani et al. 2024 |

Asli, and Senoi Orang Asli populations [15,16,32]. Our findings support the notion of the genetic flow between the Malays and the Austroasiatic-speaking Orang Aslis in Malaysia. It was presumed that the early populations of these regions might have transmigrated southward, into the Malay Peninsular, while carrying this mutation [44–47].

### 4.3 Molecular analysis using RDB-FTH and NGS

RDB-FTH assays has been utilized in multiple studies due to their technical simplicity, and time-saving [25]. A previous study employed this method to detect 17 common G6PD mutations in the Chinese population [48]. In Malaysia, previous studies have employed the Hybribio G6PDd test to detect 14 known G6PD mutations in the Malaysian population, which include variants frequently observed across all major Malaysian ethnicities, including Malays, Chinese, and Indians, thereby ensuring broad applicability of the assay across the country's major groups [25,49]. In this study, the Hybribio G6PDd test showed an excellent concordance rate of 97.4% with the NGS findings. It is consistent with those reported in past studies (94.3–100%) [48,49]. The excellent accuracy of the RDB-FTH assay, coupled with its convenience producing simultaneous results within 6 hours, indicates its high potential to be used for diagnostic services for improvement of G6PDd molecular detection. The molecular efficiency and reliability of the RDB-FTH assay can significantly strengthen the community and clinicians by enabling faster diagnostic pathology for the effective management of G6PDd.

The NGS analysis also found an additional known yet rare mutation, the rs782038151, in one subject. At the point of this research, the variant rs782038151 is available in the NCBI Single Nucleotide Polymorphism Database (dbSNP) but is not currently listed in ClinVar (link: https://www.ncbi.nlm.nih.gov/snp/rs782038151) [50]. This finding shows that, in addition to providing an unambiguous molecular diagnosis, NGS can be used to complement the PCR-based rapid detection technique such as the RDB-FTH assay, in identifying G6PDd mutations.

### 4.4 Limitations

Firstly, our study enrolled lower-than-expected number of participants from the PMOA population. This condition was most likely influenced by indigenous cultural beliefs and apprehension towards modern medicine. Our sample size may limited the number of identified cases with G6PD-deficient and G6PD-intermediate phenotypes. We also acknowledge that our sampling is small to moderate in number and may not cover all regions inhabited by the PMOA in Malaysia. Nonetheless, we believe that our study captures the major Proto-Malay settlements and provides an important first insight into the molecular and epidemiological characteristics of G6PD deficiency within this indigenous subgroup. A larger sample size could be recommended for future studies to capture more G6PD-deficient cases.

Secondly, a key limitation of this study is that only a subset of participants underwent genetic testing, primarily due to resource constraints. As such, its limited sample size and statistical power constrain the strength of the conclusions drawn from the genetic component, and the findings should be interpreted with caution. In future studies, comprehensive testing of all samples using cost-effective genotyping platforms, such as the Hybribio assay, would allow broader coverage and improve the accuracy of variant detection. Furthermore, the reliance on phenotypic enzyme activity testing alone may underestimate the number of G6PDd heterozygous females, leading to possible under-recognition of genetic diversity. Future studies should therefore prioritize wider implementation of genetic testing alongside biochemical assays.

## 5. Conclusion

This study confirmed that the PMOA group exhibits a high estimated G6PDd prevalence, along with significant molecular heterogeneity in G6PD mutations. Additionally, the molecular findings demonstrated a high concordance rate for the RDB-FTH method, with no evidence of novel G6PD mutations. These results can be interpreted as an important first step in assessing G6PDd within the PMOA population. Prioritizing G6PD screening is essential to ensure the safe administration of antimalarial drugs, particularly in this remote and vulnerable population.

## Supporting information

**S1 File. Levels of G6PD activity among Malaysia's Proto-Malay Orang Asli Indigenous population.** The study's minimal underlying G6PD activity data set.
(PDF)

**S2 File. Detected mutations and allelic frequencies in the Malaysian Proto Malay Orang Asli population.** The study's minimal underlying genetic dataset, presented as a consolidated table of all detected mutations with corresponding allele frequencies, including variants in intronic and untranslated regions.
(PDF)

**S3 File. Completed inclusivity in global research questionnaire.**
(PDF)

## Acknowledgments

We express our sincere gratitude to the Faculty of Medicine, Universiti Kebangsaan Malaysia (UKM), and the Department of Orang Asli Development (JKOA) for granting the necessary approvals to carry out this research. We also extend heartfelt thanks to all the research participants, co-investigators, and laboratory staff for their dedication and invaluable contributions to this project.

## Author contributions

**Conceptualization:** Mohamed Afiq Hidayat Zailani, Hafiza Alauddin, Ainoon Othman.

**Data curation:** Mohamed Afiq Hidayat Zailani, Muhammad-Redha Abdullah-Zawawi, Ainoon Othman.

**Formal analysis:** Mohamed Afiq Hidayat Zailani, Raja Zahratul Azma Raja Sabudin, Muhammad-Redha Abdullah-Zawawi, Najiah Ajlaa Ayub, Rinie Awai@Albert, Muhammad Dinie Afif Jamaludin, Muhammad Ziqrill Mohd Zapri, Ainoon Othman.

**Funding acquisition:** Raja Zahratul Azma Raja Sabudin.

**Investigation:** Mohamed Afiq Hidayat Zailani, Muhammad-Redha Abdullah-Zawawi, Najiah Ajlaa Ayub, Rinie Awai@Albert, Muhammad Dinie Afif Jamaludin, Muhammad Ziqrill Mohd Zapri.

**Methodology:** Mohamed Afiq Hidayat Zailani, Raja Zahratul Azma Raja Sabudin, Najiah Ajlaa Ayub, Rinie Awai@Albert, Muhammad Dinie Afif Jamaludin, Muhammad Ziqrill Mohd Zapri, Ainoon Othman.

**Resources:** Siti Aishah Sulaiman.

**Software:** Muhammad-Redha Abdullah-Zawawi.

**Supervision:** Raja Zahratul Azma Raja Sabudin, Hafiza Alauddin, Ainoon Othman.

**Validation:** Mohamed Afiq Hidayat Zailani, Muhammad-Redha Abdullah-Zawawi, Hafiza Alauddin, Ainoon Othman.

**Writing – original draft:** Mohamed Afiq Hidayat Zailani.

**Writing – review & editing:** Raja Zahratul Azma Raja Sabudin, Muhammad-Redha Abdullah-Zawawi, Azlin Ithnin, Hafiza Alauddin, Ainoon Othman.

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
