## [Decision Letter · Decision Letter 0]

15 Jul 2025

Dear Dr. Raja Sabudin,

Thank you for submitting your manuscript to PLOS ONE. After careful consideration, we feel that it has merit but does not fully meet PLOS ONE’s publication criteria as it currently stands. Therefore, we invite you to submit a revised version of the manuscript that addresses the points raised during the review process.

Please address the reviewers' comments.

We look forward to receiving your revised manuscript.

Kind regards,

Germana Bancone, Ph.D

Academic Editor

PLOS ONE

Additional Editor Comments (if provided):

Reviewers' comments:

Reviewer's Responses to Questions

**Comments to the Author**

1. Is the manuscript technically sound, and do the data support the conclusions?

Reviewer #1: Yes

Reviewer #2: Partly

2. Has the statistical analysis been performed appropriately and rigorously?

Reviewer #1: Yes

Reviewer #2: No

3. Have the authors made all data underlying the findings in their manuscript fully available?

Reviewer #1: Yes

Reviewer #2: No

4. Is the manuscript presented in an intelligible fashion and written in standard English?

Reviewer #1: Yes

Reviewer #2: No

Reviewer #1: Dear Authors,

This is a very interesting finding and you presented it beautifully. However I do have some concerns:

1. Your sample size is not very big to represent the total population of Proto-Malay Orang Asli and thus perhaps proportion is a better choice of word than prevalence in this case. Also, why sampling PMOA residing in Selangor, Negeri Sembilan and that of east Pahang only? Are these the regions where PMOA mostly reside?

2. Your concordance rate between Hybribio and NGS to be pretty high but it seems to be high only for Malaysians. I wonder whether the mutations covered in Hybribio also can detect G6PD variants common in the Chinese and Indian populations in Malaysia?

3. I see that you are still using the old WHO enzyme classification. There is already a new WHO G6PD enzyme classification : chrome-extension://efaidnbmnnnibpcajpcglclefindmkaj/https://cdn.who.int/media/docs/default-source/malaria/mpac-documentation/mpag-mar2022-session2-technical-consultation-g6pd-classification.pdf. Please do use this.

4. I see that the authors are using prevalence (proportion) for those <80% AMM as deficient. Why did you do this?Normally one used <30% as deficient and the shift drives the proportion to be higher. Lines s393-395, did the comparison with the Senoi's Orang Asli group use the same cut off ie. <80% AMM?

Reviewer #2: Manuscript: G6PD Deficiency in Malaysia’s Proto-Malay Orang Asli Indigenous Population: A Molecular and Epidemiological Study

The abstract is overly long and lacks clear structure, making it difficult to follow. Please consider shortening the content and organizing it more effectively.

The introduction is also lengthy and lacks a clear focus. Although the authors mention that the Proto-Malay Orang Asli are the second largest indigenous group in Peninsular Malaysia with a high malaria prevalence, the supporting data cited is outdated (from 2009). Furthermore, it was mentioned that P. falciparum is the predominant malaria species, but did not clearly justify why this specific population was chosen for study. The rationale for focusing on this group should be explicitly stated. Additionally, the authors should clarify the current antimalarial treatment regimen used in Malaysia, particularly whether 8-aminoquinolines (which are relevant to G6PD deficiency) are routinely prescribed. If not, the link between G6PD deficiency and clinical relevance in this population may be weak. Strengthening the rationale and ensuring it aligns with current epidemiological and treatment practices would improve the clarity and relevance of the introduction.

The manuscript would greatly benefit from a clear diagram summarizing the study design. While a total of 258 individuals were included, several essential details are missing or unclear. Specifically, the authors should indicate:

• The number of males and females included,

• How many individuals were subjected to enzyme activity testing,

• How many were excluded following family-based analysis,

• How many were tested using the Hybribio assay, and

• How many underwent next-generation sequencing (NGS).

It is also important to clarify the criteria used to select samples for Hybribio testing and NGS. On what basis were these subsets chosen?

Given the relatively small total sample size (n=258), and the fact that fewer than half underwent genetic testing, the strength of the conclusions drawn from these data is limited. It is recommended all samples should undergo genetic testing. If this is not feasible for both Hybribio and NGS platforms, at minimum, comprehensive genotyping via the simpler and more cost-effective Hybribio method should be conducted to ensure broader coverage.

Additionally, the use of only phenotypic enzyme activity tests can significantly underestimate the number of heterozygous females, as many may present with intermediate or even normal activity levels. This limitation should be clearly acknowledged, and its potential impact on data interpretation discussed.

The manuscript should address whether a sample size of 258 is statistically sufficient, especially considering only 73 samples were subjected to the Hybribio test and 39 to NGS. This limited genetic testing may reduce the reliability of the allele frequency estimates and the associations drawn.

Were any males with normal G6PD activity found to carry missense, synonymous (e.g., C1311T), or intronic mutations (e.g., T93C)? If so, these individuals should be excluded from the adjusted male median (AMM) activity calculations, as they were not wild type.

The prevalence of G6PD deficiency should be calculated using the standard threshold of 30% enzyme activity, not 80%, as the latter significantly overestimates the number of deficient individuals. Activity levels between 30–80% is classified as intermediate, and this classification should be applied consistently throughout the manuscript.

Lines 189–194, it is unclear how the variants are ordered. If they are ranked by frequency, this should be stated explicitly. Otherwise, ordering them by nucleotide position would improve clarity.

Figure 1 is difficult to interpret due to the insufficiently detailed figure legend. The caption should explain clearly how the genotyping results are presented and interpreted.

The variant rs782038151 appears unidentifiable in public databases. Please verify and correct this identifier or clarify its origin and validation.

Detailed genetic data from both Hybribio testing and NGS should be included in supplementary tables. The current presentation in the main text is difficult to follow. A consolidated table presenting all detected mutations and their corresponding allele frequencies would improve clarity. Additionally, variants in intronic and untranslated regions should be included.

The discussion is overly lengthy and should be more focused. In particular, the authors should elaborate on the implications of their findings for malaria treatment strategies, especially in the context of G6PD screening prior to the administration of 8-aminoquinolines (e.g., primaquine). The potential for implementing G6PD newborn screening in Malaysia and how this study contributes to that goal should also be discussed in greater detail.

Please note that the G6PD Viangchan variant should be correctly referred to as 871G>A—this should be corrected wherever it appears in the manuscript.

It should be clearly stated whether the G6PD Viangchan variant (871G>A) was detected in conjunction with the C1311T synonymous mutation and the T93C intronic variant. These variants are often co-inherited as part of specific haplotypes, and their co-occurrence has important implications for genotype interpretation, especially in populations with high linkage disequilibrium.

Additionally, the manuscript should be carefully revised for scientific writing. Several sections would benefit from improved clarity, conciseness, and structure.

**Do you want your identity to be public for this peer review?** For information about this choice, including consent withdrawal, please see our Privacy Policy

Reviewer #1: No

Reviewer #2: No

---

## [Author Response · Author response to Decision Letter 1]

8 Sep 2025

REVIEWER 1:

1. Thank you for your thoughtful comment. We acknowledge that our sample size is moderate in number. Nevertheless, the term prevalence is more widely used in epidemiological studies of G6PD deficiency, including those conducted on specific subpopulations, as long as the limitations are addressed. For greater precision, we have revised the manuscript to use “estimated prevalence”. Additionally, we have emphasized this point in our Methods and Discussion sections.

Regarding the sampling, we selected PMOA residing in Selangor, Negeri Sembilan, and eastern Pahang because these states and regions constitute the main settlement areas of the Proto-Malay subtribes, according to the Department of Orang Asli Development (JAKOA) demographic records. Logistical feasibility and accessibility of the communities also guided our sampling. We have added this point in our Methods, as well as our Limitations as such “We acknowledge that our sampling is small to moderate in number and may not cover all regions inhabited by the PMOA in Malaysia. Nonetheless, we believe that our study captures the major Proto-Malay settlements and provides an important first insight into the molecular and epidemiological characteristics of G6PD deficiency within this indigenous subgroup.”

2.We thank the reviewer for this insightful comment. Indeed, the Hybribio kit was designed to detect the majority of G6PD mutations reported in Malaysia, including those prevalent among the Malay, Chinese, and Indian populations. This point has been addressed in our Discussion section with reference [48], where we cited the earlier development and validation of the kit by our local researcher. To further improve clarity, we have now emphasized this in the revised text as follows “In Malaysia, the method was employed in previous studies using the Hybribio G6PDd test to detect 14 common and known G6PD mutations in the Malaysian population [25, 48], which include variants that are frequently all major Malaysian ethnicity, which include variants frequently observed across all major Malaysian ethnicities, including Malays, Chinese, and Indians, thereby ensuring broad applicability of the assay across the country’s major groups.”

3.We thank the reviewer for pointing this out. We acknowledge that the World Health Organization has updated its classification of G6PD enzyme activity (WHO, 2022). In response, we have revised our manuscript to incorporate the new classification system, replacing the old WHO categories with the updated framework. The relevant changes have been applied throughout the manuscript, particularly in Table 5, to ensure consistency. In addition, we have added the recommended WHO reference to our reference list (number 35).

4. We thank the reviewer for this important comment. We have amended our manuscript and reported the estimated prevalence at <30% as follows: “At the 30% cut-off threshold, the overall estimated prevalence of G6PDd was 6.8% (16/237; 12 males and 4 females). A total of 21 subjects were G6PD-intermediate (7 males and 14 females), and the remaining 221 subjects were G6PD-normal (72 males and 150 females).”

With regard to the comparison with the Senoi Orang Asli group (lines 393–395), with have amended our manuscript and compare our prevalence using the similar cut-off threshold of 30% of AMM, as follows:

“This study found that, at a 30% cut-off threshold, the estimated prevalence of G6PD deficiency among the PMOA population was 6.8% (16/237). This prevalence is lower than that reported in Malaysia’s Senoi Orang Asli group (9.8%; 36/369) at the same cut-off threshold using the same quantitative G6PD assay method.”

REVIEWER 2:

1. We thank the reviewer for this constructive comment. In the revised manuscript, we have substantially shortened the abstract from 303 words to 229 words. We have reorganized it to improve clarity, focus, and readability.

2. We thank the reviewer for the helpful suggestions.

a) We have shortened the introduction from 822 to 630 words. We have strengthened our supporting data on malaria prevalence by including findings from 2025 with appropriate references. " In 2025, a study of 437 Orang Asli reported a 15.3% prevalence of malaria, with Plasmodium vivax being the most common species (8.7%, 38 of 437)[14].”

Reference: Abdull-Majid, N., Yap, N. J., Tee, M. Z., Er, Y. X., Ngui, R., & Lim, Y. A. (2025). Evidence of Submicroscopic Malaria Parasitemia, Soil-Transmitted Helminths, and Their Coinfections Among Forest-Fringed Orang Asli Communities in Peninsular Malaysia. The American Journal of Tropical Medicine and Hygiene, 112(6), 1391-1399. Retrieved Aug 20, 2025, from https://doi.org/10.4269/ajtmh.24-0718

b) We have clarified the rationale for selecting this specific population in our manuscript, in the last paragraph of our Introduction section. “To date, no studies on G6PDd have been conducted among the Malaysian Proto-Malay Orang Asli (PMOA) group. This study, therefore, aims to determine the estimated prevalence of G6PDd and its molecular variants within the PMOA population. Understanding G6PDd across different Malaysian ethnicities is crucial for reducing the national disease burden and supporting policymakers in developing effective strategies for malaria eradication.”

c) We have revised the introduction to clarify the current antimalarial treatment regimen in Malaysia. “In Malaysia, Primaquine, an 8-aminoquinoline, is prescribed for the radical cure of Plasmodium vivax and as a gametocytocidal agent. As Primaquine can trigger hemolysis in G6PD-deficient individuals, determining its prevalence is clinically important to ensure safe malaria treatment.”

3.We thank the reviewer for this constructive suggestion.

a) We have added a diagram summarizing the study design (Figure 1). Kindly check the newly added figure file.

The figure indicates the total number of individuals included (n=258), with a breakdown by gender (male, n=91; female, n=167), the number subjected to enzyme activity testing (n=258),, the number excluded after family-based analysis (n=21), the number tested using the Hybribio assay(n=73) and next-generation sequencing (n=39).

b) We have clarified the inclusion criteria in the Materials and Methods section (subsection Molecular Analysis) as follows,“Following the molecular analysis by the Hybribio G6PDd test, a total of 39 samples were further sequenced via targeted sequencing (Next-Generation Sequencing; NGS). The inclusion criteria for these samples include the following: the sample that represents each type of G6PD mutation variant identified by the Hybribio G6PDd test, samples with G6PD activities <80% of the AMM that showed negative results when tested with the Hybribio G6PDd test, samples of females heterozygotes, and a subset of random G6PD-normal samples (>80% of the AMM). The selected samples were required to fulfill one or more of these criteria.”

4. We sincerely thank the reviewer for this important comment. We fully acknowledge the limitation of our study regarding the relatively small sample size and the proportion of participants who underwent genetic testing. In line with the reviewer’s suggestion, we have revised our manuscript in the Discussion section (4.4 Limitations) as follows:

“ Secondly, a key limitation of this study is that only a subset of participants underwent genetic testing, primarily due to resource constraints. As such, its limited sample size and statistical power constrain the strength of the conclusions drawn from the genetic component, and the findings should be interpreted with caution. In future studies, comprehensive testing of all samples using cost-effective genotyping platforms, such as the Hybribio assay, would allow broader coverage and improve the accuracy of variant detection. Furthermore, the reliance on phenotypic enzyme activity testing alone may underestimate the number of G6PDd heterozygous females, leading to possible under-recognition of genetic diversity. Future studies should therefore prioritize wider implementation of genetic testing alongside biochemical assays.”

5. We acknowledge the limitations of our study regarding the relatively small sample size and its statistical power. We have addressed this point in our limitations as above.

Our AMM was defined using a phenotypic approach commonly applied in most previous G6PD deficiency studies. Male participants with very low activity were excluded using a 10% activity threshold, and the median of the remaining males was taken as the reference (100%) for activity normalization. Therefore, under this definition, genotypic filtering is not required. We have clarified this in the Methods as follows: “Consistent with this AMM definition, we did not genotype-filter males with normal enzyme activity (including those carrying synonymous or intronic variants such as C1311T or T93C). [17]”

Reference: Ley B, Bancone G, von Seidlein L, Thriemer K, Richards JS, Domingo GJ, Price RN. Methods for the field evaluation of quantitative G6PD diagnostics: a review. Malar J. 2017;16(1):361. doi:10.1186/s12936-017-2017-

6. We thank the reviewer for this important comment. We have amended our manuscript and reported the estimated prevalence at <30% as follows: “At the 30% cut-off threshold, the overall estimated prevalence of G6PDd was 6.8% (16/237; 12 males and 4 females). A total of 21 subjects were G6PD-intermediate (7 males and 14 females), and the remaining 221 subjects were G6PD-normal (72 males and 150 females).” We confirmed that we have applied the corresponding classification consistently throughout the manuscript.

7. Thank you for your comment. We have arranged the reported G6PD variants by nucleotide position to improve clarity, as follows: “…a developed panel of 14 known G6PD mutation variants in Malaysia, when arranged according to their nucleotide positions, include: G6PD Gaohe (95 A>G), G6PD Orissa (131 C>G), G6PD Vanua Lava (383 T>C), G6PD Quing Yang (392 G>T), G6PD Mahidol (487 G>A), G6PD Mediterranean (563 C>T), G6PD Coimbra (592 C>T), G6PD Viangchan (871 G>A), G6PD Chatham (1003 G>A), G6PD Chinese-5 (1024 C>T), G6PD Union (1360 C>T), G6PD Andalus (1361 G>A), G6PD Canton (1376 G>T), and G6PD Kaiping (1388 G>T).”

8. Thank you for this informative comment. Figure 1 (now renamed as Figure 2) has been updated to include a revised legend, along with appropriate figure labels and title.

9. Thank you for your comment. We have verified that rs782038151 variant is available in dbSNP but is not currently listed in ClinVar (link: https://www.ncbi.nlm.nih.gov/snp/rs782038151). We have updated the manuscript to clarify its database source and ensured that the correct reference (dbSNP) is cited.

10. Thank you for this valuable suggestion. We have prepared consolidated supplementary tables that now include genetic results from both molecular tests, with intronic and untranslated regions.

11. Thank you for this suggestion. In the revised manuscript, we have streamlined the discussion and emphasized the clinical and public health implications of our findings.

12. We thank the reviewer for pointing this out. In the revised manuscript, we have corrected the nomenclature of the G6PD Viangchan variant to “871G>A” throughout the text, figures, and tables. We have also clarified the co-occurrence of the 871G>A variant with the C1311T synonymous mutation and the T93C intronic variant.

---

## [Editor Report · Decision Letter 1]

23 Sep 2025

G6PD Deficiency in Malaysia’s Proto-Malay Orang Asli Indigenous Population: A Molecular and Epidemiological Study

PONE-D-25-30027R1

Dear Dr. Raja Sabudin,

We’re pleased to inform you that your manuscript has been judged scientifically suitable for publication and will be formally accepted for publication once it meets all outstanding technical requirements.

Kind regards,

Germana Bancone, Ph.D

Academic Editor

PLOS ONE
---

## [Editor Report · Acceptance letter]

PONE-D-25-30027R1

PLOS ONE

Dear Dr. Raja Sabudin,

I'm pleased to inform you that your manuscript has been deemed suitable for publication in PLOS ONE. Congratulations! Your manuscript is now being handed over to our production team.

Kind regards,

on behalf of

Dr. Germana Bancone

Academic Editor

PLOS ONE